# Token Bottleneck: One Token to Remember Dynamics

**Taekyung Kim**[1]    **Dongyoon Han**[1]    **Byeongho Heo**[1]    **Jeongeun Park**[2]    **Sangdoo Yun**[1]

[1]NAVER AI Lab    [2]Korea University

{taekyung.k, dongyoon.han, bh.heo, sangdoo.yun}@navercorp.com    baro0906@korea.ac.kr

## Abstract

Deriving compact and temporally aware visual representations from dynamic scenes is essential for successful execution of sequential scene understanding tasks such as visual tracking and robotic manipulation. In this paper, we introduce Token Bottleneck (ToBo), a simple yet intuitive self-supervised learning pipeline that squeezes a scene into a bottleneck token and predicts the subsequent scene using minimal patches as hints. The ToBo pipeline facilitates the learning of sequential scene representations by conservatively encoding the reference scene into a compact bottleneck token during the squeeze step. In the reconstruction step, we guide the model to capture temporal dynamics by predicting the target scene using the bottleneck token along with few target patches as hints. This design encourages the vision backbone to embed temporal dependencies, thereby enabling understanding of dynamic transitions across scenes. Extensive experiments in diverse sequential tasks, including video label propagation and robot manipulation in simulated environments demonstrate the superiority of ToBo over baselines. Moreover, deploying our pre-trained model on physical robots confirms its robustness and effectiveness in real-world environments. We further validate the scalability of ToBo across different model scales. Code is available at https://github.com/naver-ai/tobo.

## 1   Introduction

With the increasing interest in deploying machines in real-world environments, ensuring seamless perception and interaction with their surroundings has emerged a crucial challenge. These operations are inherently sequential in nature, requiring the ability to trace objects (e.g., visual tracking) and predict future actions (e.g., manipulation) based on current and immediate past observations. Such understanding of the surrounding environments primarily depends on vision backbones. Therefore, a strong and robust backbone capable of generalizing across diverse tasks and environments is essential for effective sequential scene understanding.

Self-supervised learning (SSL) of visual representations has been highlighted as pivotal research in vision domains, with the pre-trained models being widely adopted for effective backbone deployment. A series of studies have introduced promising recipes for learning image [3, 4, 5, 6, 8, 14, 18] and video representations [36, 44] without labeled data. However, these studies primarily focus on understanding entire scenes or videos, which poses limitations for sequential scene understanding, as it requires capturing temporal changes across consecutive scenes and conservatively encoding the visual states of observed scenes.

To address the challenges, a sequence of studies [12, 16, 22] have attempted to incorporate correspondence learning into the MAE [18] framework, aiming to retain its strong localization capability while enabling the model to match corresponding regions across consecutive scenes. However, we observe that such additional considerations have a limited impact on the quality of scene representations and may result in suboptimal performance in sequential scene understanding tasks, such as robotic manipulation (§3.2). This limitation arises since recognizing temporal changes alone is insufficient;

39th Conference on Neural Information Processing Systems (NeurIPS 2025).

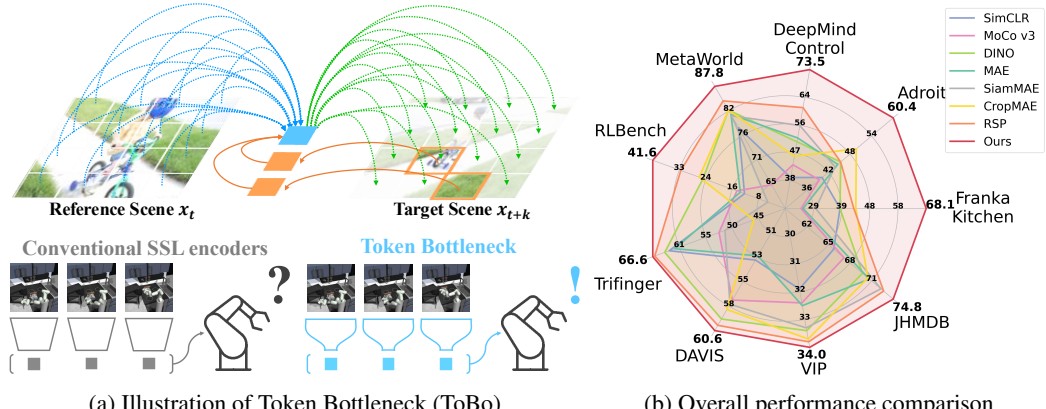

(a) Illustration of Token Bottleneck (ToBo)  (b) Overall performance comparison

Figure 1: (a) We describe the underlying mechanism of our **Token Bottleneck (ToBo)** pipeline during pre-training, which conservatively encode a reference scene into a bottleneck token and predict the subsequent target scene based on a scarce target patches and the bottleneck token. ToBo facilitates learning the capability of temporal progression recognition and preservation of observed information (top). Therefore, using bottleneck tokens from the current and recent past observations enables the robot to better understand its current state (bottom). (b) Our method significantly surpasses previous self-supervised visual representation learning methods designed for static [4, 5, 8, 18] and dynamic scenes [16, 22, 49] on various robot manipulation and locomotion tasks.

these tasks require the ability to summarize the essential information from each scene without loss, while preserving temporal cues within the summarized representation.

In this paper, we introduce Token Bottleneck (ToBo), a simple yet effective SSL approach that intuitively facilitates the conservative summarization of observed scenes while enabling effective recognition of temporal evolution within the summarized representations. As illustrated in Fig. 1a, ToBo squeezes a reference scene into a bottleneck token and then predicts the subsequent target scene using only a minimal set of patches as hints. This design enforces strong reliance on the bottleneck token, encouraging the vision backbone to capture essential scene information. Moreover, predicting the target scene from the bottleneck token implicitly embeds temporal dependencies, guiding the vision backbone to generate representations capable of capturing dynamic transitions across consecutive scenes.

We conduct comprehensive experiments to assess the effectiveness of our pre-training pipeline in comparison with existing self-supervised learning methods. We evaluate our method on various sequential understanding tasks, including manipulation tasks in simulated environments and video label propagation tasks, surpassing baselines [4, 5, 8, 12, 16, 18, 22, 49] with significant gaps (see Fig. 1b). Furthermore, we deploy our pre-trained models on real-world robots, demonstrating strong generalization performance in unseen physical environments. Finally, we validate the scalability of our approach by observing consistent performance gains across various model scales.

## 2 Related Work

**Self-supervised learning on a static scene.**   Self-supervised learning (SSL) approaches have been widely explored in the image domain. Contrastive learning approaches [3, 5, 7, 8, 17] aim to learn useful representations by maximizing the similarity between positive pairs derived from a static scene through strong augmentations. Although these methods excel in facilitating a cohesive understanding of images, they suffer from limited localization capabilities [27], essential for action prediction in robotics. On the other hand, masked image modeling (MIM) [1, 2, 18, 27, 28, 50] has recently gained attention for its promising capacity to learn visual representations through predictive learning. Inspired by masked language modeling (MLM) in transformers [10], BEiT [2] extends MLM into the vision domain, adopting an external offline tokenizer. MAE [18] and SimMIM [50] showcase efficient MIM by directly reconstructing masked input pixels without any tokenizer. However, these approaches do not incorporate mechanisms for capturing temporal progression during pre-training.

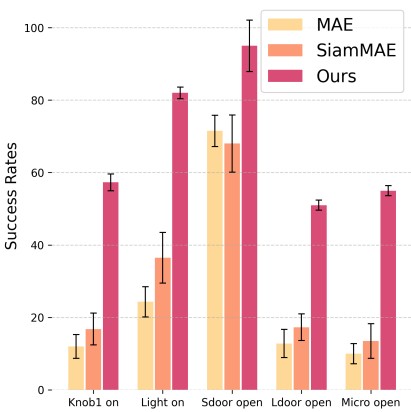

Figure 2: **Comparative analysis for motivation.** We compare robot manipulation performance using MAE and SiamMAE as visual backbones. While SiamMAE employ temporal correspondence to the limitation of MAE, its improvement over MAE remains limited.

Figure 3: **Overview of our Token Bottleneck (ToBo).** Our ToBo reconstructs the masked patches from the *bottleneck token* representation of the reference scene $\mathbf{x}^t$ and extremely scarce patches from the target scene $\mathbf{x}^{t+k}$. Such extreme scarcity leads the decoder $d_\phi$ to rely heavily on the reference scene $\mathbf{x}^t$, facilitating the preservation of observed information in the *bottleneck token*.

**Self-supervised learning on dynamic scenes.** Recent studies have focused on enhancing the recognition of dynamic transitions. SiamMAE [16] proposes visual representation learning methods that utilize dynamic scenes. CropMAE [12] introduce a simple augmentation strategy that enables the generation of dynamic scenes even from a single static image. On the other hand, RSP employs stochastic frame prediction tasks along with masked autoencoding. Several works have also explored applying these techniques to embodied agents and robotic manipulation. For example, VC-1 [33], MVP [40], and Dasari et al. [9] adopt MAE objectives for visual pretraining, while STP [51] builds on SiamMAE with a reference masking strategy. On the other hand, some prior works investigate representation learning with annotated supervision. Theia [42] distills representations from large scale pre-trained teacher networks, some of which are trained with annotation supervision, into student models. MPI [24], Voltron [25], and R3m [34] explore language-driven representation learning, leveraging an auxiliary textual guidance through manually annotated data. In contrast, we focus on self-supervised learning directly from raw dynamic scenes without any guidance from annotations.

## 3 Method

### 3.1 Preliminary

**Masked autoencoding.** Given a scene image, we patchify the image into $N$ non-overlapping $p \times p$-size patches $\{\mathbf{x}_i\}_{i=1}^N$ where $\mathbf{x}_i \in \mathbb{R}^{3p^2}$. We randomly select a masked patch set $\mathcal{M} \subset \{1, 2, ..., N\}$ with a ratio $r \in (0, 1)$ where $|\mathcal{M}| = \lfloor rN \rfloor$. The remaining patches $\{\mathbf{x}_i\}_{i \in \mathcal{M}^c}$ fed into the encoder $f_\theta$, becoming spatial representations $\{\mathbf{u}_i\}_{i \in \mathcal{M}^c}$ where $\mathbf{u}_i \in \mathbb{R}^d$ for encoder dimension $d$. Note that a learnable CLS token $e_{[CLS]}$ is also encoded with spatial representations as a part of the encoding process. The encoded tokens are reconstructed to $N$ tokens by substituting masked positions to a mask token $\mathbf{m} \in \mathbb{R}^d$. i.e. $\mathbf{u}_i \leftarrow \mathbf{m}$ for $i \in \mathcal{M}$. The decoder $d_\phi$ gets $\{\mathbf{u}_i\}_{i=1}^N$ as input and predicts the masked image patches $\{\hat{\mathbf{x}}_i\}_{i \in \mathcal{M}}$ using encoded tokens.

### 3.2 Motivation

Real-world robots must see and act seamlessly in complex dynamic environments. This fundamental challenge motivates our central research question: how should a vision backbone learn to capture spatiotemporal relationships across a sequence of observations? While conventional self-supervised vision encoders [16, 18, 22, 50] have been widely employed for this purpose, they possess significant limitations from the perspective of sequential scene understanding. In this section, we discuss the strengths and limitations of these prior approaches, thereby providing the insights for our proposed method.

**Limits of self-supervised learning on static scenes**  Self-supervised learning approaches on static scenes such as MAE [18] and SimMIM [50] are effective for appearance modeling and localization, which leads to their adoption in several robotics studies [9, 33, 40]. These strengths stem from the design that enforces the autoencoder to predict missing information from available prior information (e.g., visible patches). This pipeline implicitly encourages the encoder to facilitate interactions among the remaining sparse tokens, thereby enhancing localization capabilities. However, since their predictive learning is performed exclusively within single static scenes, the encoder is never explicitly optimized to compare consecutive frames, leaving them ineffective at modeling temporal dynamics. Moreover, a recent study reveals that they struggle with learning broader contexts [27], leading to representations with a limited cohesive understanding. These limitations further constrain their potential for sequential scenes understanding. Consequently, MAE shows limited sequential understanding and thus underperforms some manipulation task, as depicted in Fig. 2.

**Limits of self-supervised learning with patch-wise temporal correspondence**  To alleviate the chronic limitations of static scene-based SSL approaches, SiamMAE [16] builds a non-trivial correspondence matching problem by randomly sampling two dynamic scenes from sequential data. The core principle involves propagating patches from the reference scene to their corresponding locations in the target scene. Applying this guidance using a cross-attention layer-based decoder encourages patch-level correlation between target patches and reference patches. However, while the training objective successfully enforces patch-level correspondences, it overlooks the crucial step of interpreting what these collective matches represent from a holistic perspective. Consequently, despite being built upon the MAE framework, its performance gain over MAE is marginal or even negative in some sequential scene-based tasks, as shown in Fig. 2. This suggests that fine-grained recognition of temporal evolution is insufficient for sequential scene understanding, and a conservative summarization of the observed scenes is essential.

**Computational inefficiency of combinatorial architectures**  A common approach to achieve comprehensive capabilities is to construct a combinatorial architecture that integrates separate pipelines specialized for each desired capability. For example, RSP [22] combines masked autoencoding, global representation alignment, and target scene reconstruction for localization, global understanding, and patch-level correlation, respectively. However, this combinatory design leads to a substantial increase in computational overhead. As a result, RSP requires more than double the computation cost of competing methods, as reported in Table 9.

### 3.3   The Proposed Method - Token Bottleneck (ToBo)

**Our claim**  Our goal is to achieve representations optimized for resolving sequential scene-based tasks. In light of the discussions in §3.2, we extend our focus beyond simply recognizing temporal evolution; we consider the conservative summarization of observed scenes in a way that also effectively embeds temporal dynamics within the summarized representation.

To this end, we present Token Bottleneck (ToBo), a self-supervised visual representation learning pipeline that enables these capabilities through a token bottleneck mechanism. ToBo consists of two key steps: squeezing a scene into a single token, which we denote as the *bottleneck token*, and reconstructing information from this token. Suppose a reference scene and a target scene are given. In the squeeze step, visual information from the reference scene is compactly encoded into the bottleneck token. Subsequently, in the reconstruction step, we guide the model to predict the target scene using the bottleneck token, with only a minimal set of patches from the target scene provided as hints. In this situation, the model cannot precisely reconstruct the target scene based solely on the limited hints, which strengthens the reliance of the reconstruction step on the bottleneck token. This design yields two advantages: (1) the bottleneck token should preserve essential information from the reference scene, and (2) such information should be encoded in a way that enables recognition of temporal dynamics when interleaved with the hints from the target scene. Eventually, our goal can be achieved by optimizing the objective of the Token Bottleneck pipeline. The overall description of our pipeline is depicted in Fig. 3.

**Overall pipeline formulation**  Suppose we sample a reference scene $\mathbf{x}^t \in \mathbb{R}^{3 \times H \times W}$ and a target scene $\mathbf{x}^{t+k} \in \mathbb{R}^{3 \times H \times W}$ with a temporal gap $k$. We patchify $\mathbf{x}^t$ and $\mathbf{x}^{t+k}$ into $N$ non-overlapping patches $\{\mathbf{x}_i^t\}_{i=1}^N$ and $\{\mathbf{x}_i^{t+k}\}_{i=1}^N$, respectively. The reference scene patches $\{\mathbf{x}_i^t\}_{i=1}^N$ are fed into an encoder $f_\theta$, yielding spatial representations $\{\mathbf{u}_i^t\}_{i \in \mathcal{M}^c}$. We use the CLS token output from this encoding process as the bottleneck token $\mathbf{u}_{tobo}$, which will be guided to compactly summarize the

Table 1: **Experimental results on vision-based robot policy learning on Franka Kitchen.** We report the performance of imitation learning agents on Franka Kitchen [15], which are trained upon representations from the ViT-S/16 model pre-trained on Kinetics-400 [26] dataset. The success rates (%) are reported for all the tasks. We underline the second-best performance. We report the gains of our method over the second-best baseline.

| Tasks | SimCLR | MoCo v3 | DINO | MAE | SiamMAE | RSP | CropMAE | ToBo |
|---|---|---|---|---|---|---|---|---|
| Knob1 on | $25.3\pm2.1$ | $11.5\pm3.9$ | $27.0\pm3.2$ | $12.0\pm3.3$ | $16.8\pm4.4$ | $31.0\pm2.4$ | $\underline{31.5}\pm5.3$ | $\mathbf{57.0}\pm2.0$ |
| Light on | $\underline{55.8}\pm6.4$ | $24.3\pm5.0$ | $44.3\pm6.5$ | $24.3\pm4.2$ | $36.5\pm7.0$ | $44.5\pm5.6$ | $54.0\pm11.2$ | $\mathbf{82.0}\pm1.6$ |
| Sdoor open | $72.3\pm2.8$ | $66.5\pm3.2$ | $77.0\pm5.0$ | $71.5\pm4.3$ | $68.0\pm7.9$ | $\underline{82.5}\pm2.7$ | $77.0\pm8.1$ | $\mathbf{95.0}\pm7.1$ |
| Ldoor open | $17.0\pm2.9$ | $10.3\pm2.1$ | $16.5\pm2.5$ | $12.8\pm3.9$ | $17.3\pm3.7$ | $\underline{28.8}\pm4.8$ | $25.5\pm5.7$ | $\mathbf{51.0}\pm1.4$ |
| Micro open | $23.3\pm2.8$ | $14.3\pm2.5$ | $28.5\pm4.8$ | $10.0\pm2.8$ | $13.5\pm4.8$ | $30.3\pm5.6$ | $\underline{32.5}\pm4.1$ | $\mathbf{55.0}\pm1.4$ |

Table 2: **Experimental results on vision-based robot policy learning on CortexBench.** The performance of imitation learning agents on CortexBench [33] is reported, where the agents are trained upon representations from the ViT-S/16 model pre-trained on the Kinetics-400 [26] dataset. We report the normalized score for DeepMind Control Suite (DMC) and success rates (%) for other tasks. We report the gains of our method over the second-best baseline.

| Tasks | SimCLR | MoCo v3 | DINO | MAE | SiamMAE | RSP | CropMAE | ToBo |
|---|---|---|---|---|---|---|---|---|
| Adroit | $40.4\pm3.3$ | $39.6\pm4.3$ | $45.6\pm6.2$ | $39.6\pm4.3$ | $44.0\pm6.6$ | $45.6\pm4.6$ | $\underline{50.0}\pm5.1$ | $\mathbf{60.4}\pm2.2$ |
| MetaWorld | $78.4\pm5.2$ | $65.4\pm8.0$ | $82.4\pm5.8$ | $65.4\pm8.0$ | $81.1\pm6.3$ | $\underline{84.5}\pm6.6$ | $82.4\pm5.8$ | $\mathbf{87.8}\pm4.6$ |
| DMC | $39.7\pm2.9$ | $43.7\pm3.2$ | $50.9\pm1.5$ | $43.7\pm3.2$ | $56.0\pm2.9$ | $\underline{61.6}\pm3.4$ | $46.4\pm1.1$ | $\mathbf{73.5}\pm0.9$ |
| TriFinger | $63.3\pm3.3$ | $53.3\pm1.6$ | $64.2\pm3.5$ | $53.3\pm1.6$ | $52.1\pm7.6$ | $\underline{66.2}\pm0.8$ | $46.3\pm1.7$ | $\mathbf{66.5}\pm1.0$ |

reference scene. The target scene $\{\mathbf{x}_i^{t+k}\}_{i=1}^N$ is masked with an extremely high ratio $r \in (0,1)$, where $\mathcal{M} \subset \{1,2,...,N\}$ and $|\mathcal{M}| = \lfloor rN \rfloor$. The unmasked target patches $\{\mathbf{x}_i^{t+k}\}_{i \in \mathcal{M}^c}$ are processed by the same encoder $f_\theta$, producing $\{\mathbf{u}_i^{t+k}\}_{i \in \mathcal{M}^c}$ for the target scene. We then concatenate the bottleneck token $\mathbf{u}_{tobo}$ with the target representations $\{\mathbf{u}_i^{t+k}\}_{i \in \mathcal{M}^c}$ and fill mask tokens $\mathbf{m}$ for missing regions $i \in \mathcal{M}$. These are passed to the decoder $d_\phi$, which predicts the masked image patches $\{\hat{\mathbf{x}}_i^{t+k}\}_{i \in \mathcal{M}}$ by using $\mathbf{u}_{tobo}$ and $\{\mathbf{u}_i^{t+k}\}_{i \in \mathcal{M}^c}$. Due to the extremely high masking ratio applied to the target scene, the decoder $d_\phi$ proactively rely on $\mathbf{u}_{tobo}$, which enable the encoder $f_\theta$ to conservatively summarize the reference scene in a way that facilitates temporal reasoning when compared to the target hints. We minimize the reconstruction loss throughout the training as follows:

$$\mathcal{L}_{\text{ToBo}} = \sum_{i \in \mathcal{M}} d(\hat{\mathbf{x}}_i^{t+k}, \mathbf{x}_i^{t+k}), \tag{1}$$

where $d(\cdot)$ is a distance function; we use cosine distance for the pre-training.

**Decoder structure** Previous methods in dynamic SSL [12, 16, 22] utilize cross-attention layers as a core component for learning temporal evolution awareness, placing them within the decoders to guide the encoder to learn representations that effectively capture correspondences. These approaches leverage a hybrid structure of cross-attention layers, self-attention layers, and multi-layer perceptron (MLP) layers. In contrast, ToBo employs self-attention layers to ensure that the decoder exclusively attends to the given information during the reconstruction step, with MLP layers for progressive transformation from representation embedding spaces into the pixel space.

## 4 Experiment

In this section, we focus on demonstrating the effectiveness of our pre-training pipeline through fair comparisons with existing self-supervised learning methods. To this end, we evaluate our method on sequential tasks, including video label propagation tasks [23, 38, 55] and vision-based policy learning for robotic manipulation and locomotion across various simulated environments [15, 21, 33]. We extend our validation to real-world settings by deploying our pre-trained model on physical robots, showcasing its transferability. We further investigate the scalability of our method. In the appendix, we validate our claim regarding the importance of extremely high masking ratios to the target scene, present qualitative comparisons of manipulation processes against baseline methods, and show provide demonstrations of real-world manipulation tasks.

Table 3: **Experimental results on vision-based robot policy learning on RLBench.** We report the performance of imitation learning agents on RLBench [21], which are trained upon representations from the ViT-S/16 model pre-trained on Kinetics-400 [26] dataset. The success rates (%) are reported for all the tasks. We report the gains of our method over the second-best baseline.

| Tasks | SimCLR | MoCo v3 | DINO | MAE | SiamMAE | RSP | CropMAE | ToBo |
|-------|--------|---------|------|-----|---------|-----|---------|------|
| Button | $7.4_{\pm2.6}$ | $11.4_{\pm4.1}$ | $24.7_{\pm1.5}$ | $6.4_{\pm2.2}$ | $6.1_{\pm2.3}$ | $\underline{28.4}_{\pm3.0}$ | $26.9_{\pm6.7}$ | $\mathbf{41.2}_{\pm7.4}$ |
| Phone | $34.6_{\pm6.6}$ | $36.2_{\pm3.4}$ | $32.0_{\pm5.5}$ | $37.7_{\pm1.9}$ | $5.4_{\pm0.5}$ | $\underline{48.0}_{\pm4.6}$ | $16.6_{\pm3.8}$ | $\mathbf{52.3}_{\pm3.2}$ |
| Umbrella | $5.8_{\pm3.3}$ | $13.2_{\pm1.5}$ | $28.1_{\pm1.4}$ | $10.0_{\pm1.2}$ | $4.0_{\pm0.0}$ | $37.3_{\pm3.0}$ | $\underline{37.5}_{\pm8.8}$ | $\mathbf{42.2}_{\pm6.9}$ |
| Wine | $11.0_{\pm2.1}$ | $8.7_{\pm0.7}$ | $31.4_{\pm1.5}$ | $10.0_{\pm2.1}$ | $8.7_{\pm0.8}$ | $31.9_{\pm2.3}$ | $\underline{33.2}_{\pm0.2}$ | $\mathbf{35.4}_{\pm3.8}$ |
| Rubbish | $5.2_{\pm1.2}$ | $6.7_{\pm0.8}$ | $12.9_{\pm1.5}$ | $6.2_{\pm3.2}$ | $3.5_{\pm0.9}$ | $18.5_{\pm1.1}$ | $\underline{20.6}_{\pm1.7}$ | $\mathbf{37.0}_{\pm6.1}$ |

Table 4: **Performance on real-world vision-based robot policy learning.** Success rates (%) of imitation learning agents on three manipulation tasks: Cabinet Opening, Drawer Closing, and Cup Stacking. Agents are trained with ViT-S/16 representations pre-trained on Kinetics-400 [26] for 400 epochs. The results demonstrate the generalizability of ToBo in real-world.

| Method | Cabinet Opening | Drawer Closing | Cup Stacking |
|--------|-----------------|----------------|--------------|
| SiamMAE | 20.0 | 55.0 | 50.0 |
| RSP | 25.0 | 65.0 | 55.0 |
| CropMAE | 0.0 | 25.0 | 20.0 |
| ToBo (ours) | **65.0** | **75.0** | **80.0** |

Figure 4: **Real-world robot trajectories.** Initial, intermediate, and final states of the robot during (a) Cabinet Opening, (b) Drawer Closing, and (c) Cup Stacking.

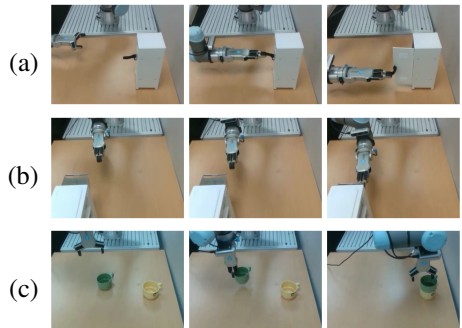

(a)

(b)

(c)

## 4.1 Experimental Setup

**Implementaion details**  We follow the evaluation protocol of Jang et. al. [22] for both video label propagation and vision-based policy learning on simulated environments. To ensure fair comparisons with the baselines, we also pre-train our method on Kinetics-400 for 400 epochs. Detailed explanation for both pre-training and evaluation are provided in the Appendix.

**Baselines**  We compare the performance of our method with conventional self-supervised learning (SSL) methods for visual representations including SimCLR [5], MoCo v3 [8], DINO [4], and MAE [18] We also consider previous dynamic scene SSL methods, i.e., SiamMAE [16], RSP [22], and CropMAE [12]. We validate the impacts of explicitly learning state representations over these approaches.

## 4.2 Vision-based robot policy learning in simulated environments

We evaluate our method through imitation learning on robot manipulation and locomotion tasks across various simulated environments. Specifically, we evaluate five tasks from both the Franka Kitchen and RLBench benchmarks. Moreover, we consider two, five, five, and two tasks from Adroit [41], MetaWorld [53], DeepMind Control Suite (DMC) [43], and TriFinger [46] from the CortexBench benchmark, respectively.

**Franka Kitchen.** We present a comparison between our method and the baselines on vision-based robot policy learning in the Franka Kitchen environment in Table 1. The results demonstrate that our method significantly outperforms all the baselines across all tasks. Notably, our method achieves over 20% improvements in success rates on all tasks, except for the *Light on* task. This highlights the effectiveness of explicitly encoding visual state representation for vision-based robot policy learning.

**CortexBench.** We compare our method with the baselines for the vision-based robot manipulation and locomotion tasks in the Adroit, MetaWorld, DeepMind Control (DMC), and Trifinger environments in Table 2. The results show that our method achieves superior performance compared to the baselines across all tasks. In particular, our method surpasses the second-best performance with success rate gains of 11.9%p on DMC and 10.4%p on Adroit.

Table 5: **Results on video label propagation.** We report performances on video segmentation, video part segmentation, and pose tracking tasks from DAVIS [38], VIP [55], and JHMDB [23] benchmarks, respectively. For all methods, we report the performance with the representations pre-trained on the Kinetics-400 [26] dataset for 400 epochs.

| Method | DAVIS | | | VIP | JHMDB | |
| --- | --- | --- | --- | --- | --- | --- |
| | $\mathcal{J}\&\mathcal{F}_m$ | $\mathcal{J}_m$ | $\mathcal{F}_m$ | mIoU | PCK@0.1 | PCK@0.2 |
| SimCLR | 53.9 | 51.7 | 56.2 | 31.9 | 37.9 | 66.1 |
| MoCo v3 | 57.7 | 54.6 | 60.8 | 32.4 | 38.4 | 67.6 |
| DINO | 59.5 | 56.5 | 62.5 | 33.4 | 41.1 | 70.3 |
| MAE | 53.5 | 50.4 | 56.7 | 32.5 | 43.0 | 71.3 |
| SiamMAE | 58.1 | 56.6 | 59.6 | 33.3 | 44.7 | 73.0 |
| RSP | 60.1 | 57.4 | 62.8 | 33.8 | 44.6 | 73.4 |
| CropMAE | 58.6 | 55.8 | 61.4 | 33.7 | 42.9 | 71.1 |
| ToBo (ours) | **60.6** | **58.4** | **63.0** | **34.0** | **47.0** | **74.8** |

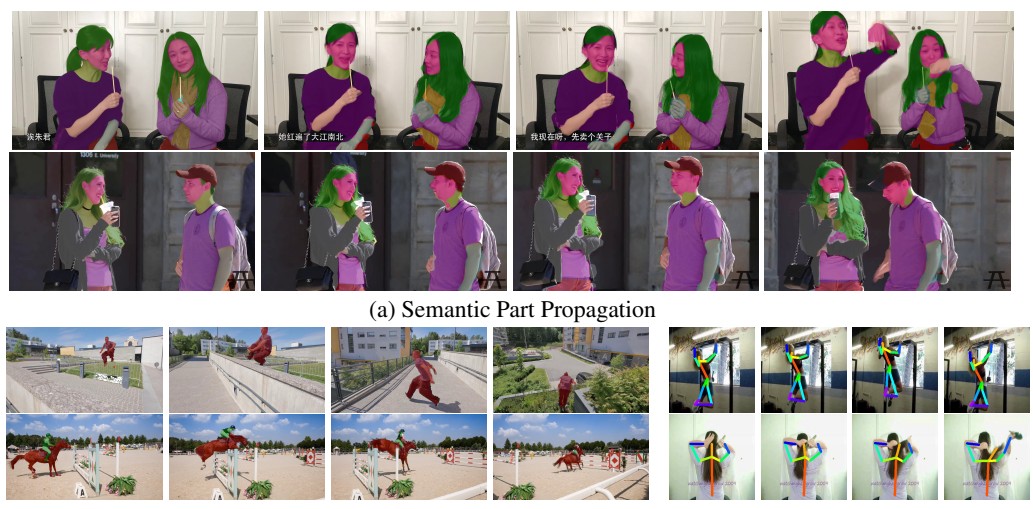

(a) Semantic Part Propagation

(b) Object Propagation        (c) Pose Tracking

Figure 5: **Qualitative results for video label propagation.** We provide examples of predicted propagation of our model on video object segmentation, video part segmentation, and pose tracking benchmarks. The leftmost images indicate the ground-truth annotations. We visualize the propagated results corresponding to 25, 50, and 100% ratio of the videos.

**RLBench.** Table 3 showcases the robot manipulation performance on five demonstration tasks in the RLBench environment. Notably, our method consistently exceeds all baselines across the five tasks. Moreover, the degraded performance of MAE and SiamMAE further highlights the significance of state representation learning for the robot backbones.

### 4.3 Vision-based Robot Policy Learning in Real-world Environments

**Quantitative comparison.** To validate the robustness of our method in real-world environments, we further investigate SSL methods on real-world robot manipulation tasks. Specifically, we design three demonstration tasks: *Cabinet Opening*, *Drawer Closing*, and *Cup Stacking*. For each task, We collect 50 demonstration episodes for training and 10 demonstration episodes for evaluation for imitation learning. Following the training protocol used in simulated environments, we train the policy network using a standard behavior cloning loss. The experimental results for each individual task are reported in Table 4. We first observe that our method exceeds SiamMAE [16], RSP [22], and CropMAE [12] on all three tasks. Specifically, our method improves 40%p, 10%p, and 25%p over the baselines on the *Cabinet Opening*, *Drawer Closing*, and *Cup Stacking* tasks, respectively. While previous SSL methods on dynamic scenes struggle with tasks that require relatively high precision, like cabinet opening tasks, our method even successfully executes the task with a considerable success rate. This showcases that models pre-trained by our method can be robustly transferred to real-world environments.

Table 6: **Comparison with robot representation learning models.** We compare the performance of our method with robot representation learning methods across multiple simulated manipulation tasks. We categorize the methods into self-supervised learning, supervised learning through foundation models, and supervised learning with auxiliary language guidance. Despite the unbalanced training and evaluation setup, ToBo surpasses the RRL models on MetaWorld. Moreover, ToBo exceeds self-supervised RRL models despite of a smaller model with a significantly smaller amount of data. These results demonstrate the effectiveness of the representations learned by ToBo in diverse robot manipulation tasks.

| Method | #Param | Dataset | #Seen frames | Adroit | MetaWorld | Franka Kitchen |
|---|---|---|---|---|---|---|
| *Supervision through Foundation Models* | | | | | | |
| Theia[†] | 52.9M | Theia dataset | 14.4B[*] | 66.0 | 86.1 | - |
| *Supervision with Auxiliary Language Guidance* | | | | | | |
| R3M | 25.6M | Ego4D | 0.8B | 65.0 | 69.2 | 53.1 |
| MVP | 21.7M | MVP dataset | 4.8B | - | 84.6 | - |
| Voltron[‡] | 21.7M | SS-v2 | 0.3B | - | 68.7 | 70.5 |
| MPI[‡] | 21.7M | Ego4D | 0.1B | - | 85.7 | 76.5 |
| *Self-supervised Learning* | | | | | | |
| R3M[°] | 25.6M | Ego4D | 0.8B | 45.6 | 67.0 | 47.2 |
| data4robotics | 86.0M | Kinetics-700 | 0.5B | - | 87.0 | 55.0 |
| VC-1 | 86.0M | Ego4D+N | 1.0B | 50.0 | 86.4 | - |
| ToBo (ours) | 21.7M | Kinetics-400 | 0.2B | 60.4 | 87.8 | 68.0 |

[†] Uses additional compression layers. [‡] Uses multi-head attention pooling layers for integrating spatial tokens.
[°] Excludes language guidance from the vanilla recipe. [*] Includes data for distillation models.

**Qualitative comparison**    To illustrate the actual manipulation processes, we present the robot trajectories from successful demonstrations for three real-world manipulation tasks in Fig. 4. Specifically, the initial states of the physical robot are depicted in the left scenes, while the right scenes show the final states of the demonstrations. The middle scenes illustrate the intermediate states of the demonstrations. Our model clearly succeeds in all the tasks. We also compared the trajectories with the baselines in the Appendix.

## 4.4 Video Label Propagation

We perform comparative analyses on the video label propagation tasks. We consider the video object segmentation, video part segmentation, and pose tracking tasks from DAVIS [38], VIP [55], and JHMDB [23]. We follow the evaluation protocol in Jang et. al [22]. We present the quantitative evaluation in Table 5. Our method demonstrates superior performance compared to all the baselines across the video label propagation tasks. We also provide qualitative results in Fig. 5, where our method effectively traces visual appearances across various video label propagation tasks. These visualizations highlight that our method maintains robust object identity, part consistency, and pose continuity. The strong performance in both quantitative and qualitative evaluations further demonstrate the effectiveness of our approach in capturing the temporal evolution of visual appearance across consecutive scenes.

## 5 Discussion

**Comparison with robot representation learning models**    We further compare our method with recent robot representation learning (RRL) models, categorized by their supervision types: self-supervised learning [9, 33], supervision via foundation model outputs [42], and supervision with auxiliary language annotations [24, 25, 34, 40]. Table 6 shows the reported performance of RRL models across several simulated robot manipulation benchmarks [15, 41, 53]. Here, our model is based on a ViT-Small architecture trained on Kinetics-400 for 400 epochs. Notably, despite having the smallest number of parameters and the second smallest amount of training data, and using no annotation-based supervision, our method achieves the highest score on MetaWorld. In particular, Theia is trained by distilling knowledge from five large-scale foundation models (CLIP large [39], Depth Anything large [52], DINOv2 large [35], Segment Anything huge [29], and ViT huge [45]),

Table 7: **Performance with vision-language models.** We compare the performance of our method with vision-language models on Franka Kitchen. Despite using a smaller model, significantly less pre-training data, and no auxiliary textual guidance from manually annotated data, ToBo consistently outperform the other models across all tasks.

| Method | #Param | Dataset | #Seen frames | Knob1 on | Light on | Sdoor open | Ldoor open | Micro open |
|---|---|---|---|---|---|---|---|---|
| CLIP* | 149.3M | WebImageText | 12.8B | 23.0 | 29.5 | 69.5 | 13.5 | 22.0 |
| DINOv2 | 22.1M | LVD-142M | 4.3B | 25.5 | 38.0 | 82.0 | 15.5 | 20.0 |
| SigLIP* | 203M | WebLI | 2.1B | 17.5 | 38.5 | 75.0 | 8.5 | 16.5 |
| SigLIP2* | 375M | WebLI | 40B | 11.0 | 23.5 | 58.5 | 11.0 | 18.0 |
| ToBo (Ours) | 21.7M | Kinetics-400 | 0.2B | **57.0** | **82.0** | **95.0** | **51.0** | **55.0** |
| Gain | | | | + 31.5 | + 43.5 | + 13.0 | + 35.5 | + 33.0 |

\* The model is trained using textual guidance with manually annotated data.

which are collectively trained on 14.3 billion annotated samples. It also employs convolution-based compression layers during evaluation. Surpassing Theia under such an unbalanced training and evaluation setup is noteworthy. Furthermore, the performance gap between R3M with and without language guidance highlights the substantial benefit of auxiliary language supervision. Even with such unfairness in the training setup, our method outperforms R3M, MVP, Voltron, and MPI on MetaWorld. It also surpasses R3M on Franka Kitchen, despite significant differences in training data and model size. Compared to self-supervised RRL models, our method outperforms all the models. It surpasses much larger models such as VC-1 and data4robotics, despite being trained on a significantly smaller amount of data. Given the minimal number of parameters and training scale, these results demonstrates the effectiveness and efficiency of our proposed method for robot manipulation tasks.

**Comparison with Vision-Language Models**  We compare our method with vision-language models widely used either as backbones across various domains or as vision towers in large language models. For fair evaluation, we follow the same evaluation protocol used in the main paper. We evaluate CLIP [39], DINOv2 [35], SigLIP [54], and SigLIP2 [47] in the Franka Kitchen benchmark, as shown in Table 7. Despite having the smallest number of learnable parameters and being exposed to the smallest number of seen frames during pre-training, ToBo achieves consistently superior performance, outperforming the baselines by margins at least 13.0%p to the maximum 43.5%p. These performance gaps are notable given that all baselines except DINOv2 use language supervision from manually annotated data. These results demonstrate the effectiveness of ToBo in summarizing visual observations for sequential scene understanding tasks.

**Ablation Study on Mask Ratio of Target Scenes**
To verify our claim that extremely scarce information from target scenes forces the decoder to rely highly on the stored visual scene information of the reference scene, we conduct an ablation study varying the mask ratio of target scenes. We pre-train the models on the Kinetics-400 [26] dataset for 100 epochs and evaluate five tasks on Franka Kitchen. As shown in Figure 6, the effectiveness of our proposed method increases as the masking ratio of target scenes increases until 0.9, verifying our claim that scarce target scene information facilitates the exploitation of the compressed reference information. Besides, the models pre-trained with a masking ratio of 0.95 yield degraded performance in some tasks, demonstrating that minimal clues are necessary for the prediction of the missing information.

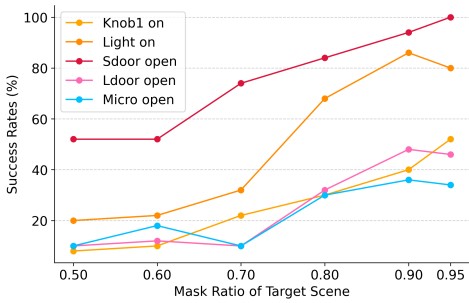

Figure 6: **Varying the masking ratio of target scenes.** We vary the masking ratio from 0.5 to 0.95 and pre-train ViT-S/16 models on the Kinetics-400 [26] dataset for 100 epochs.

**Scalability**  We investigate the scalability of our ToBo beyond ViT-S/16 by pre-training ViT-B/16 and ViT-L/16 on Kinetics-400 [26] for 100 epochs. We evaluate the pre-trained models on vision-based robot policy learning on Franka Kitchen [15] using three different seeds. We compare our method with MAE, SiamMAE, and RSP. Table 8 presents the mean and standard deviation across all seeds. We observe that models pre-trained with ToBo consistently achieving the best

Table 8: **Scalability of our method.** We report the performance of vision-based robot policy learning on Franka Kitchen [15], which are trained upon representations from the ViT-B/16 and ViT-L/16 model pre-trained on Kinetics-400 [26] dataset for 100 epochs. The success rates (%) are reported for all the tasks. We underline the second-best performance. We report the gains of our method over the second-best baseline. We conduct evaluations using three different seeds.

| Arch. | Method | Knob1 on | Light on | Sdoor open | Ldoor open | Micro open |
|---|---|---|---|---|---|---|
| ViT-B/16 | MAE | $18.7_{\pm1.2}$ | $21.3_{\pm4.6}$ | $70.0_{\pm2.0}$ | $17.3_{\pm2.3}$ | $15.3_{\pm2.3}$ |
| | SiamMAE | $18.0_{\pm2.0}$ | $34.0_{\pm2.0}$ | $80.7_{\pm3.1}$ | $18.7_{\pm1.2}$ | $19.3_{\pm6.1}$ |
| | RSP | $\underline{24.7}_{\pm3.1}$ | $\underline{51.7}_{\pm9.1}$ | $\underline{87.3}_{\pm2.3}$ | $\underline{23.3}_{\pm7.6}$ | $\underline{26.7}_{\pm2.3}$ |
| | ToBo (ours) | $\mathbf{46.7}_{\pm6.4}$ | $\mathbf{78.7}_{\pm7.6}$ | $\mathbf{95.3}_{\pm1.2}$ | $\mathbf{47.3}_{\pm5.0}$ | $\mathbf{37.3}_{\pm4.6}$ |
| | Gain | + 22.0 | + 27.0 | + 8.0 | + 24.0 | + 10.6 |
| ViT-L/16 | MAE | $19.3_{\pm7.6}$ | $33.3_{\pm2.3}$ | $61.3_{\pm6.4}$ | $16.0_{\pm2.0}$ | $14.0_{\pm2.0}$ |
| | SiamMAE | $20.7_{\pm3.1}$ | $34.0_{\pm4.0}$ | $76.0_{\pm2.0}$ | $12.7_{\pm6.4}$ | $22.0_{\pm0.0}$ |
| | RSP | $\underline{26.7}_{\pm2.3}$ | $\underline{48.0}_{\pm2.0}$ | $\underline{88.0}_{\pm2.0}$ | $\underline{22.7}_{\pm8.3}$ | $\underline{23.3}_{\pm4.2}$ |
| | ToBo (ours) | $\mathbf{54.7}_{\pm5.0}$ | $\mathbf{75.3}_{\pm4.2}$ | $\mathbf{94.0}_{\pm3.5}$ | $\mathbf{50.0}_{\pm2.0}$ | $\mathbf{42.7}_{\pm6.1}$ |
| | Gain | + 28.0 | + 27.3 | + 6.0 | + 27.3 | + 19.4 |

performance across all five tasks, exhibiting significant improvements over the second-best results. These demonstrate the scalability of our method.

**Comparison of training and inference flops** We conducted FLOPs evaluation for both training and inference to quantitatively compare the computational cost of each model, as summarized in Table 9. During inference, all models use the same backbone architecture and input resolution without any input masking, resulting in identical inference FLOPs at the same model scale (e.g., 4.6 GFLOPs for ViT-Small). During training, ToBo, MAE [18], and SiamMAE [16] show similar com-

Table 9: Comparison of training FLOPs and downstream performance in Franka Kitchen.

| Method | Training FLOPs (GFLOPs) | Franka Kitchen (%) |
|---|---|---|
| MAE | 13.0 | 26.1 |
| SiamMAE | 13.1 | 30.4 |
| RSP | 32.5 | 43.4 |
| ToBo | 15.9 | **68.1** |

putational costs while RSP [22] requires substantially more computation of 32.5 GFLOPs due to its complex decoding mechanisms. When considering computational costs with downstream performance (e.g., performance in Franka Kitchen [15]), these results further support the effectiveness of ToBo, which achieves a strong balance between efficiency and performance.

## 6 Conclusion

We have introduced Token Bottleneck (ToBo), a self-supervised visual representation learning method designed for sequential scene understanding. The backbones for sequential scene-based tasks should effectively preserve visual information from observations while facilitating the recognition of temporal progression across sequential scenes. While conventional self-supervised learning (SSL) methods have proven promising impacts in visual representation learning, they primarily focus on understanding static images or entire videos, often lacking embeds for handling dynamic transitions in sequential tasks. Recent SSLs aim to address this by adapting correspondence learning in dynamic scenes. However, their patch-wise representations of observations are often suboptima for subsequent policy networks, especially in tasks like robotic manipulation. To this end, ToBo introduces a simple yet effective pipeline that facilitates conservative summarization of the observed scene into a bottleneck token while enable capturing of dynamic transitions through the bottleneck token. Through extensive experiments in various sequential understanding tasks including manipulation tasks and video label propagation tasks, we verified the superiority of ToBo over conventional SSL methods and previous dynamic scene SSL methods. Furthermore, applying ToBo in real-world settings demonstrates its robustness and generalization capability.

**Limitation** Due to the resource constrains, we did not check the scalability of our method beyond huge scale and explore beyond the commonly used input resolution. Additionally, our study focused on a simplest setting involving two dynamic scenes to learn temporal dynamics.

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

## Appendix

## A    Further Analysis

In this section, we examine how hyperparameter choices in the Token Bottleneck (ToBo) pre-training pipeline affect performance, regarding the number of bottleneck tokens, impact of temporal difference, We pre-train ViT-B/16 for 100 epochs on Kinetics-400 [26] throughout the ablation studies. The comparisons are done on five Franka Kitchen imitation-learning tasks [15].

**Ablation study on the number of bottleneck tokens**    We vary the number of bottleneck tokens in {1, 2, 4, 8}. As shown in Table Aa, using a single token consistently yields the best performance across tasks. This demonstrates that conservative summarization without auxiliary storage better captures the current observation and thus improves action prediction in robotics.

**Ablation study on temporal difference**    We further vary the maximum temporal gap between frames among {48, 96, 144}. As shown in Table Ab, moderate temporal differences encourage the model to learn dynamic scene evolution, since shorter gaps lack meaningful change whereas overly longer gaps disrupt temporal coherence

**Ablation study with no temporal difference**    We apply our method using the same frame for both source and target scenes. As shown in Table Ac, our method still works even without temporal difference, surpassing other baselines (e.g., MAE [18], SiamMAE [16], and RSP [22]) with significant margin across tasks. However, its overall performance degrades compared to original ToBo, reflecting the loss of supervision from temporal change. This highlights the importance of temporal contrast for effective pre-training of ToBo.

**Ablation study on multiple source frames**    We compare ToBo to a variant pre-trained with multiple source frames. Specifically, we randomly sample four source frames and pre-train for 100 epochs under the same recipe as ToBo. As shown in Table Ad, this multi-frame variant surpasses prior baselines (e.g., MAE [18], SiamMAE [16], and RSP [22]) in most of the tasks. However, despite requiring higher pre-training cost, it underperforms compared to ToBo across all robotics tasks. These results suggest that while it is possible to extend ToBo to multi-frame settings, such naive extension may encounter potential new challenges, leading to suboptimal performance.

## B    Manipulation trajectory visualization of Real-world Demonstrations

We showcase the robot manipulation trajectories for the SiamMAE [16], RSP [22], and our model as robot backbones in the same episode on the real-world environment for each task. In Figure A, Specifically, the leftmost scenes depicts the initial states of the physical robot, while the rightmost scenes show the final states of the demonstrations. As shown in Fig. A, while SiamMAE and RSP fail to execute the manipulation tasks, our method successfully completes them within the same episode. We also provide videos of these demonstrations in the supplementary material.

## C    Implementation Details

We provide implementation details for pre-training and evaluation. Specifically, we present the evaluation protocols for vision-based robot policy learning on each simulated environment (i.e., Franka Kitchen [15], CortexBench [33], RLBench [21]) and real-world environment. Then, we explain experimental setups for video label propagation tasks.

Table A: **Ablation studies Token Bottleneck pre-training**. We vary hyperparameters of the Token Bottleneck (ToBo) pre-training pipeline. We report the success rates (%) on five imitation learning tasks from the Franka Kitchen benchmark [15]. All models are ViT-B/16 and are pre-trained for 100 epochs on Kinetics-400 [26]. We mark our default settings in  gray .

(a) **Number of bottleneck tokens**

| # bottleneck tokens | Knob1 on | Light on | Sdoor open | Ldoor open | Micro open | Mean |
|---|---|---|---|---|---|---|
| 1 | **46.7** | **78.7** | **95.3** | **47.3** | **37.3** | **61.1** |
| 2 | 31.0 | 54.0 | 74.0 | 26.0 | 24.0 | 41.8 |
| 4 | 28.0 | 24.3 | 78.0 | 28.0 | 22.0 | 36.1 |
| 8 | 10.0 | 20.0 | 56.0 | 26.0 | 9.3 | 24.3 |

(b) **Temporal difference**

| Maximum temporal difference | Knob1 on | Light on | Sdoor open | Ldoor open | Micro open | Mean |
|---|---|---|---|---|---|---|
| 48 | 40.7 | 78.7 | 96.0 | 44.0 | 35.3 | 58.9 |
| 96 | **46.7** | **78.7** | 95.3 | **47.3** | 37.3 | **61.1** |
| 144 | 36.0 | 69.3 | **97.3** | 46.7 | **39.3** | 57.7 |

(c) **Ablation with no temporal difference**

| Method | Knob1 on | Light on | Sdoor open | Ldoor open | Micro open | Mean |
|---|---|---|---|---|---|---|
| MAE [18] | 18.7 | 21.3 | 70.0 | 17.3 | 15.3 | 28.5 |
| SiamMAE [16] | 18.0 | 34.0 | 80.7 | 18.7 | 19.3 | 34.1 |
| RSP [22] | 24.7 | 51.7 | 87.3 | 23.3 | 26.7 | 42.7 |
| ToBo (no temporal difference) | 41.0 | 72.0 | 89.3 | 32.7 | 32.0 | 53.5 |
| ToBo | 46.7 | 78.7 | 95.3 | 47.3 | 37.3 | **61.1** |

(d) **Ablation on multiple source frames**

| Method | Knob1 on | Light on | Sdoor open | Ldoor open | Micro open | Mean |
|---|---|---|---|---|---|---|
| MAE [18] | 18.7 | 21.3 | 70.0 | 17.3 | 15.3 | 28.5 |
| SiamMAE [16] | 18.0 | 34.0 | 80.7 | 18.7 | 19.3 | 34.1 |
| RSP [22] | 24.7 | 51.7 | 87.3 | 23.3 | 26.7 | 42.7 |
| ToBo (w/ multi-frame) | 28.7 | 60.7 | 92.7 | 20.7 | 32.0 | 46.9 |
| ToBo | 46.7 | 78.7 | 95.3 | 47.3 | 37.3 | **61.1** |

## C.1 Pre-training

We pre-train ViT-S/16 [11] on Kinetics-400 [26] for 400 epochs for the main comparison, while we pre-train ViT-S/16, ViT-B/16, and ViT-L/16 for 100 epochs for analyses. We employ repeated sampling [19, 13] with a factor of 2 so that the models are indeed pre-trained for 200 epochs. We use AdamW optimizer [32] with a batch size of 1536, comprising dynamic scenes with a resolution of 224×224. These scenes are randomly sampled from videos at a rate of 30 FPS, with a temporal index gap ranging from 4 to 96. We simply apply random resized crop and horizontal flip to the scenes, aligning the cropping region across the reference and target scenes. To drive the learning mechanism of our proposed method, we randomly mask the target scenes with an extremely high masking ratio of 0.9. Our decoder is composed of eight vision transformer blocks, i.e., each block contains self-attention layers and multi-layer perceptrons. We follow the default hyperparameters of the baselines for their pre-training on Kinetics-400 [26]. We adopt a siamese masked autoencoding loss [16] as an auxiliary objective to enhance learn patch-level correspondence learning.

## C.2 Vision-based Robot Policy Learning

**Franka Kitchen.** We validate models pre-trained by our method and other baselines in five imitation learning tasks from the Franka Kitchen benchmark [15]. Our experiments mainly follow the imitation learning evaluation setup in Jang et. al. [22], which builds upon [34, 37]. Specifically, we employ an agent comprising a frozen backbone initialized with pre-trained models and a policy network consisting of a two-layer MLP, with a batch normalization layer applied at the input stage. We define the state representation for the policy network as the combination of the visual representation and the robot's proprioception. For the perception, we employ either a left or right camera with a 224×224 resolution while omitting depth. The policy network is trained with a standard behavior cloning loss. Training for each demonstration task progresses for 20,000 steps, with a periodic online

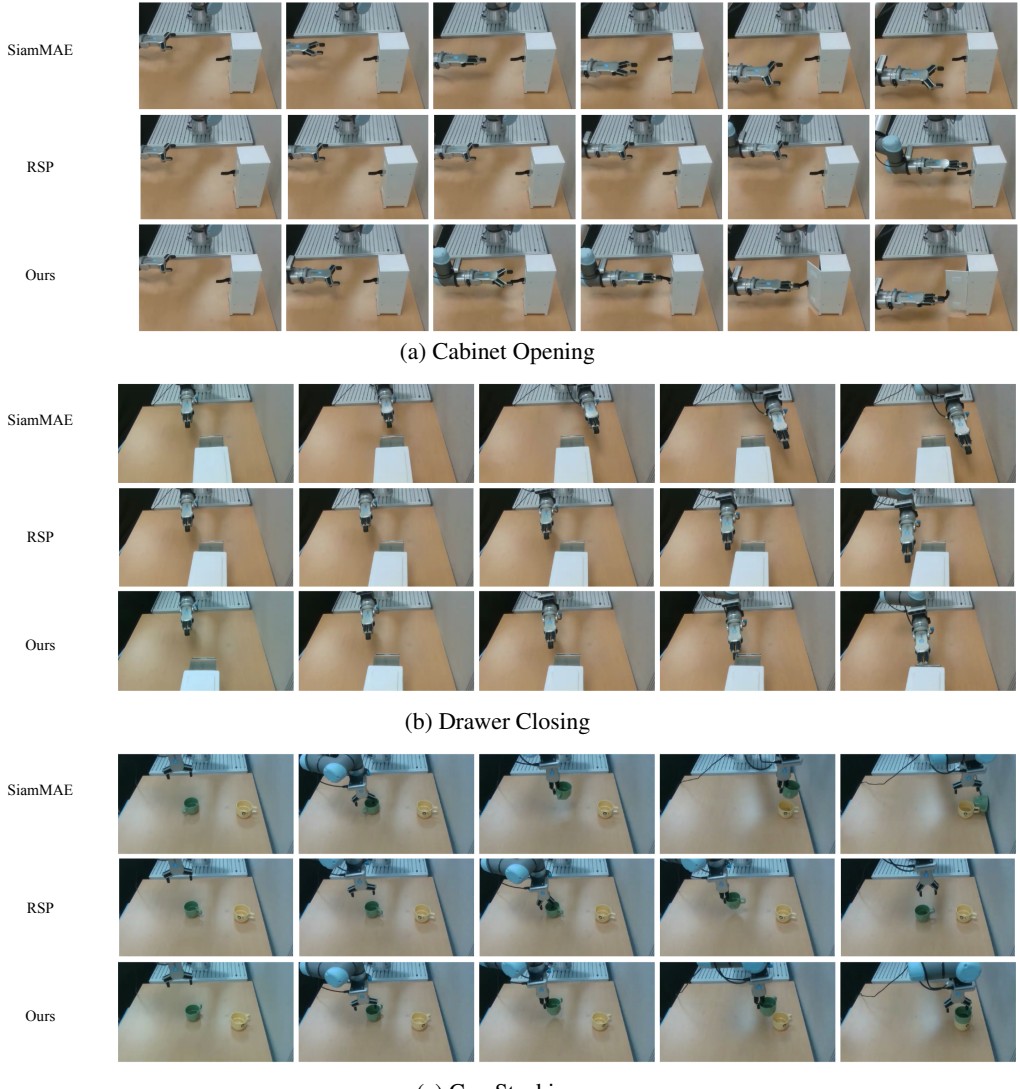

(a) Cabinet Opening

(b) Drawer Closing

(c) Cup Stacking

Figure A: **Sampled Trajectories from Real World Experiment**. We visualize the manipulation trajectories of ToBo, RSP, and SiamMAE on physical robot manipulation tasks in real-world environments (i.e., *Cabinet Opening*, *Drawer Closing*, and *Cup Stacking*). Our ToBo successfully demonstrates all tasks, which aligns with the quantitative performance comparisons results.

evaluation in the simulated environment every 1,000 steps. We evaluate the highest success rates of each demonstration across four different seeds and report its average with a 95% confidence interval.

**RLBench.** We consider five manipulation tasks from RLBench [21]. Follow the evaluation setup in Jang et. al. [22], we generate 100 demonstrations and utilize them for training the agent. We employ a front camera with a 224×224 resolution. Point cloud information is excluded throughout all experiments. We employ the end-effector controller with path planning. We evaluate the highest success rates of each demonstration across four different seeds.

**CortexBench.** We evaluate the models on four simulated environments from CortexBench [33]. We consider two, five, five, and two demonstrations from Adroit, DeepMind Control (DMC) [43], Meta-World [53], and Trifinger, respectively. Proprioceptive data is utilized except the DMC benchmark. We mainly follow the experimental setups in Jang et. al. [22], which builds upon [33]. For each task, we train the agent for 100 epochs, with a periodic online evaluation in the simulated environment

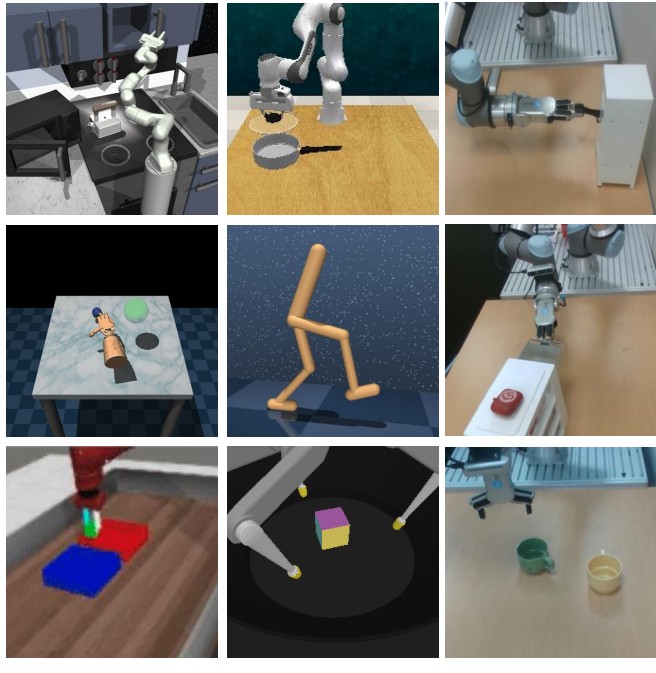

|  (a) Simulated environments | (b) Real-world |

Figure B: **Visualization of environments for robot policy learning evaluation.** We validate the effectiveness of our method on (a) simulated environments (e.g., Franka Kitchen [15], CortexBench [33], RLBench [21] and (b) real-world environments. We design real-world environments with physical robots to evaluate how the algorithm handles given tasks.

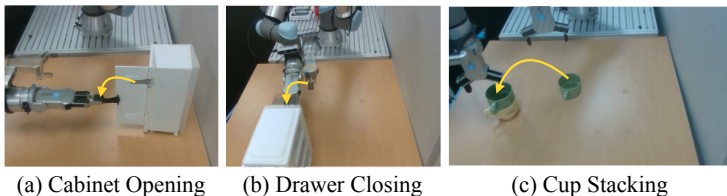

| (a) Cabinet Opening | (b) Drawer Closing | (c) Cup Stacking |

Figure C: **Task Description for Real-world Environments.** We illustrates the objectives of physical robot manipulation tasks in the real-world. Yellow arrows indicate the target actions for each task.

every 5 epochs. We report the normalized score for DMC and the highest success rates for other tasks. We conduct demonstration tasks for five different seeds and report its average with a 95% confidence interval.

**Real-world Environments.** We evaluate our proposed method in real-world robotic imitation learning tasks using a UR5e manipulator equipped with a parallel gripper. The policy operates at a control frequency of 5 Hz, executing actions defined as delta end-effector poses and gripper's state, with specific parameterizations for each task: (dx, dy) for drawer closing, (dx, dy, gripper open/close) for cabinet opening, and (dx, dy, dz, gripper) for cup stacking. The system employs joint position control at 50 Hz, with a numerical inverse kinematics (IK) solver running in the background to calculate the end-effector's pose to the joint position. Our training dataset consists of 50 demonstrations for cabinet opening and cup stacking and 30 demonstrations for drawer closing. We train the two-layer MLP policy for 100 epochs without incorporating proprioceptive states, using a top-front camera view with a resolution of 224×224. The final performance is evaluated based on the reported average success rate across tasks. Figure C provides visual examples of the three tasks under consideration.

**Video label propagation.** We conduct comparative analyses for video label propagation on video object segmentation on DAVIS [38], video part segmentation on VIP [55], and pose tracking on

JHMDB [23]. Following the evaluation protocols in the previous studies [48, 31, 30, 20], we employ $k$-nearest neighbor inference, maintain a queue of length $m$ to provide temporal context, and restrict the set of source nodes within a spatial radius $r$. Additionally, we perform a grid search to optimize evaluation hyperparameters for each method and report the best results.

