# OpenReview forum: "Token Bottleneck: One Token to Remember Dynamics"
_NeurIPS.cc/2025/Conference — NeurIPS 2025 poster_

### Official Review · Reviewer_3Edo · 2025-06-22

**Clarity:** 3
**Significance:** 2
**Originality:** 2
**Rating:** 4
**Confidence:** 4

**Summary:**

This work depicts that a single token is sufficient to compress the spatial and temporal dynamics of a video. Specifically, this work demonstrates that utilizing the class token of the reference frame in conjunction with a few target patches facilitates the learning of sequential scene representations.

**Questions:**

See weaknesses

**Ethical Concerns:**

["NO or VERY MINOR ethics concerns only"]

**Final Justification:**

The authors have addressed many of my concerns and I believe that this paper can be accepted. So I’m maintaining my rating.

**Limitations:**

yes

**Quality:**

3

**Strengths And Weaknesses:**

Strengths:-

- The paper is predominantly well-written.
- Extensive experiments on various downstream tasks such as robot policy learning have been conducted.
- The main insight presented in the paper that utilizing a single token from the reference frame instead of utilizing all the patches is more beneficial, is very interesting.

Weakness:-

- The decoder architecture is not clear. While the visualization in figure 3 indicates attention from the bottleneck token to the unmasked patches, it is mentioned in L-156 that self-attention is employed. A better description would be helpful.
- While the experiments are predominantly comprehensive, more comparisons with works performing masked image modeling on latent space such as i-BOT, DINOv2 and T-LORE could have been conducted.

---

> ### Author Rebuttal · Authors · 2025-07-31
>
> We sincerely thank Reviewer 3Edo for the thoughtful comments and constructive suggestions. Below, we provide detailed responses to each point.
>
> > ### **W1. Clarification on the description of the decoder architecture**
>
> We apologize for the confusion. During the decoding process, the visible and masked target patches are concatenated with the bottleneck token and then fed into the self-attention layer. We promise to address the confusions in the descriptions.
>
> > ### **W2. Comparison with i-BOT, DINOv2 and T-CORE**
>
> We provide the comparison results of DINOv2 [11], iBoT [15], and T-CORE [16] in five tasks in Franka Kitchen. Since we have provided the performance of DINOv2 in Franka Kitchen in the supplementary material (Table A), we directly report the results in **Table R13**. Moreover, we further evaluate the performance of iBoT and T-LORE. As shown in **Table R13**, our ToBo surpasses DINOv2, iBoT, and T-LORE on all the five tasks. We promise to provide the performance of DINOv2, iBoT, T-LORE in other benchmarks in the revised version.
>
>
> [**Table R13.** Comparison of ToBo with DINOv2 and T-CORE on Franka Kitchen]
>
>
>
> |                             |    Knob1 on   |   Light on    |   Sdoor open    |   Ldoor open    |   Micro open    |
> |------------------------|------------------|-----------------|---------------------|---------------------|---------------------|
> | DINOv2         |         27.3     |        38.7      |         82.0        |        16.0           |        20.0         |
> | iBoT             |         37.3    |        55.3      |         75.3          |        12.7          |        47.3          |
> | T-CORE         |       7.3        |        17.3      |         69.3        |          11.3         |       8.0             |
> | |
> | ToBo (ours)          |        **57.3**      |        **82.0**      |        **95.0**          |        **51.0**          |        **55.0**         |
>
>
>
> ### **References**
>
> [11] Oquab et. al., "DINOv2: Learning Robust Visual Features without Supervision", arXiv:2304.07193.
>
> [15] Zhou et., al., iBOT: Image BERT Pre-Training with Online Tokenizer, ICLR 2022.
>
> [16] Liu et. al., "When the Future Becomes the Past: Taming Temporal Correspondence for Self-supervised Video Representation Learning", CVPR 2025.

---

> > ### Comment · Reviewer_3Edo · 2025-08-03
> >
> > I would like to thank the authors for their rebuttal. All of my concerns have been well addressed. However, I do think that other reviewers have raised valid points and would like to wait for their opinions on the rebuttals before making a decision on my final rating.

---

> > > ### Comment · Area_Chair_NAu8 · 2025-08-05
> > >
> > > Dear reviewer 3Edo,
> > >
> > > We appreciate your timely response to authors. We encourage you to update the score nevertheless, based on your own assessment, since the concerns of other reviewers will be factored in their own scores already. The authors have posted the rebuttal to other reviewers, if you still would like to take them into account.
> > >
> > > -- Your AC

---

> > > > ### Comment · Reviewer_3Edo · 2025-08-05
> > > >
> > > > I would like to maintain my previous positive rating.

---

> > > > > ### Author Response · Authors · 2025-08-06
> > > > >
> > > > > Thank you again for your valuable comments! We're glad that your concerns have been addressed. Please feel free to let us know if there are any remaining points we can further clarify!

---

### Official Review · Reviewer_dztw · 2025-07-01

**Clarity:** 2
**Significance:** 2
**Originality:** 2
**Rating:** 4
**Confidence:** 3

**Summary:**

The authors propose a self-supervised vision pretraining method for sequential scene based tasks by bottlenecking the representation using a single token. The authors demonstrate the effectiveness of this approach in simulation and real world robotics tasks, with significant improvements over other methods like SimCLR and MAE(and its variants). The authors also demonstrate superior performance of their proposed method in label(mask,pose) propagation tasks over these baselines.

**Questions:**

see weaknesses

**Ethical Concerns:**

["NO or VERY MINOR ethics concerns only"]

**Final Justification:**

The proposed SSL method beats SoTA methods like I/V-JEPA, DinoV2 on robotics tasks and can scale with increase in training data.

**Limitations:**

yes

**Quality:**

2

**Strengths And Weaknesses:**

Strengths
1. The authors propose a novel self-supervised pretraining method to capture both temporal dynamics and summarize scene information in the same embedding.
2.  The authors demonstrate the superior performance of their proposed method in robotics tasks and label propagation when compared to other SSL methods like MAE variants and SimCLR.



Weaknesses
1. The authors do not compare with newer self supervised vision pretraining methods like DINO v2 (except franka-kitchen) and I/V-JEPA on all the tasks. This is required to contextualise the  performance of the proposed method in the current SSL landscape.
2. The authors need to demonstrate the scaling property of the method, when trained on more data and not just a larger model.
3. The improvements in label propagation are much smaller than the improvements in robotics policy learning tasks. This needs to be explained/investigated. Is this an artefact of the agent's policy architecture being more suited to the proposed SSL method as opposed to the other SSL methods? Is the agent policy architecture the same for all the robotics tasks ?
4. While the authors focus on bottlenecking the representation using a single embedding, it would be interesting to see the effect of increasing the number of embeddings in the bottleneck effectively relaxing the constraint gradually.

---

> ### Author Rebuttal · Authors · 2025-07-31
>
> We sincerely thank Reviewer dztw for the thoughtful comments and constructive suggestions. Below, we provide detailed responses to each point.
>
> > ### **W1. Comparison with DINO v2 (beyond franka-kitchen), I-JEPA, and V-JEPA**
>
> Following the comments, we provide the comparison results of DINOv2 [11], I-JEPA [12], V-JEPA [13]. We provide the performance of I-JEPA and V-JEPA for five tasks of Franka Kitchen in **Table R10**. Moreover, since we have provided the performance of DINOv2 in Franka Kitchen in the supplementary material (Table A), we additionally compare ToBo with DINOv2 on 'relocate' and 'pen' tasks in Adroit in **Table R11**. As shown in **Table R10**, our ToBo surpass DINOv2, I-JEPA, and V-JEPA on all the tasks in Franka Kitchen. Moreover, ToBo also outperforms DINOv2 on all the tasks in Adroit. These results further support the effectiveness of ToBo. We promise to provide the performance of DINOv2, I-JEPA, V-JEPA in other benchmarks in the revised version.
>
>
> [**Table R10.** Comparison of ToBo with DINOv2, I-JEPA, and V-JEPA on Franka Kitchen]
>
> |                             |    Knob1 on   |   Light on    |   Sdoor open    |   Ldoor open    |   Micro open    |
> |------------------------|------------------|-----------------|---------------------|---------------------|---------------------|
> | DINOv2        |         27.3     |        38.7      |         82.0        |        16.0           |        20.0         |
> | I-JEPA           |         26.0     |        40.0      |         78.7        |         21.3         |         17.3         |
> | V-JEPA          |        21.3      |        41.3      |         74.7        |         28.0          |         12.7          |
> | |
> | ToBo (ours)         |        **46.7**      |        **78.7**       |        **95.3**          |        **47.3**          |        **37.3**       |
>
>
> [**Table R11.** Comparison of ToBo and DINOv2 on Adroit]
>
> |                             |   Pen    |   Relocate    |
> |------------------------|-----------|------------------|
> | DINOv2         |  69.6    |       24.0       |
> | ToBo (ours)         |  **81.6**    |       **39.2**       |
>
>
>
>
>
>
> > ### **W2. Scalability of our method when trained on more samples**
>
> Following the advice, we verify the scalability of ToBo when trained on more samples. To this end, we compare ToBo models pre-trained on Kinetics-400 and Kinetics-600, a larger dataset that includes Kinetics-400, for 100 epochs. **Table R12** showcases that the ToBo model pre-trained on Kinetics600 surpasses the ToBo model pre-trained on Kinetics-400in all tasks except for the "Sdoor Open" task, where both models perform comparably. This results validate the scalability of ToBo for more samples.
>
>
> [**Table R12.** Scalability of our method when trained on more samples]
>
> |                             |    Knob1 on   |   Light on    |   Sdoor open    |   Ldoor open    |   Micro open    |
> |------------------------|------------------|-----------------|---------------------|---------------------|---------------------|
> | ToBo pre-trained on Kinetics-400   |        46.7      |        78.7       |        **95.3**          |        47.3          |        37.3       |
> | ToBo pre-trained on Kinetics-600 |        **51.0**     |        **80.3**      |        94.7          |        **48.7**          |        **45.3**       |
>
>
> > ### **W3. Clarification for much smaller improvements on label propagation**
>
> The improvements in video label propagation are smaller because it requires dense and spatially distributed predictions, which do not directly align with our SSL objective. In contrast, the strong gains in robotics stems from the nature of our SSL objective, which enforces conservative summarization of observed scenes into a bottleneck token, a representation better suited for action prediction under partial observability.
>
>
> The following are responses to the sub-questions:
>
>
> **Response to "Is the agent's policy architecture being more suited to the proposed SSL method?"**
> The observed improvements in robotics tasks are not due to any special advantage in policy architecture. Our method directly adopts the policy network design and the evaluation protocol from prior works [3], without any modification specific to our SSL objective. Importantly, all compared SSL methods share the same policy architecture, ensuring a fair comparison. Therefore, the gains are not attributable to an architectural bias toward our method.
>
> **Response to "Is the agent policy architecture the same for all the robotics tasks?"**
> Yes, the same policy network architecture is used across all robotics tasks. This uniform setup eliminates architectural variation as a source of performance gaps. Although the training mechanisms of models vary across SSL methods, all models share the same structure, which enables the use of a single policy design across the pre-trained models.
>
>
>
>
>
> ### **References**
>
> [1] He et. al., "Masked Autoencoders are Scalable Vision Learners," CVPR 2022.
>
> [2] Gupta et. al., "Siamese masked autoencoders," NeurIPS 2023.
>
> [3] Jang et. al., "Visual representation learning with stochastic frame prediction", ICML 2024.
>
> [6] Kay et. al., "The kinetics human action video dataset," arXiv:1705.06950, 2017.
>
> [11] Oquab et. al., "DINOv2: Learning Robust Visual Features without Supervision", arXiv:2304.07193.
>
> [12] Bardes et. al., "Revisiting Feature Prediction for Learning Visual Representations from Video", arXiv:2404.08471.
>
> [13] Assran et. al., "Self-Supervised Learning from Images with a Joint-Embedding Predictive Architecture," ICCV 2023.

---

> > ### Comment · Reviewer_dztw · 2025-08-03
> >
> > I would like to thank the authors for their response. My concerns have been addressed.

---

> ### Comment · Area_Chair_NAu8 · 2025-08-05
>
> Thank you for responding to authors. Would you mind to do the "Mandatory Acknowledgement" for the review? Did the author's response change your score?
>
> -- Your AC

---

> > ### Comment · Reviewer_dztw · 2025-08-06
> >
> > I would like to maintain my positive rating.

---

> ### Author Response · Authors · 2025-08-06
>
> Thank you for the thoughtful feedback! We're glad that our response addressed your concerns. Please feel free to let us know if there are any remaining points we can further clarify!

---

### Official Review · Reviewer_GhiU · 2025-07-03

**Clarity:** 3
**Significance:** 3
**Originality:** 3
**Rating:** 4
**Confidence:** 3

**Summary:**

This paper investigates how to use the dynamics information in video to pretrain an image encoder, extending MAE. The proposed ToBo(Token Bottleneck) has a unique pretext task of compressing an image into a single token, then decoding a later frame based on the bottleneck token and sparse target frame patches. The method is mainly evaluated on video label propagation and robotics tasks, yielding especially large improvements in robotics manipulation.

**Questions:**

Please see the weakness section. To summarize, I have questions regarding the variance of performances, more careful discussion of limitations, and direct video pretraining. I am open to increasing the score if these items are clarified.

**Ethical Concerns:**

["NO or VERY MINOR ethics concerns only"]

**Final Justification:**

I believe this paper makes good progress on representation learning and it is meaningful to the community. My final rating is borderline accept.

**Limitations:**

The limitation discussion is too short, probably due to paper length limitation. But it should be revisited more carefully.

**Paper Formatting Concerns:**

None.

**Quality:**

3

**Strengths And Weaknesses:**

Strength:
1. The proposed approach is simple but effective. The pretext task intuitively makes sense and is grounded in previous literature.
2. The improvements on especially robotics tasks is impressive.
3. Robotics experiments include error bars, which is informative.

Weaknesses:
1. The variance on the proposed approach seems especially large, especially in Table 3, compared to the baselines. Could the author provide some insight into why that is the case?
2. More careful discussion of the limitation should be included in the final paper.
3. Since we are learning representation from video, is it possible to take the VideoMAE approach to pretrain a video model, then somehow adapt it to the robotics use case?

---

> ### Author Rebuttal · Authors · 2025-07-31
>
> We sincerely thank Reviewer GhiU for the thoughtful comments and constructive suggestions. Below, we provide detailed responses to each point.
>
> > ### **W1. Clarification on concerns regarding variance**
>
> * While addressing your comment, we found that the variance of ToBo in the DMC benchmark [7] was incorrectly reported as 8.6, when it should have been 0.86. We will correct this error (**8.6 --> 0.9**) in Table 2 of the main paper, as reflected in **Table R7**.
> * ToBo may tend to exhibit relatively high variance in RLBench [10]. However, we believe this reflects a broader trend where model variances vary significantly depending on the evaluation benchmark. In fact, on other benchmarks such as CortexBench [8] (**Table R7**) and Franka Kitchen [4] (**Table R8**), ToBo shows lower or comparable variance than other methods except “Sdoor Open”. Moreover, even in the "Sdoor Open" task, ToBo records a variance of 7.1, which is lower than that of SiamMAE (7.9) and CropMAE (8.1). These results suggest that variance is task-dependent, and ToBo maintains stable performance in several benchmarks despite relatively higher sensitivity in certain RLBench scenarios
>
>
> [**Table R7.** Experimental results on vision-based robot policy learning on CortexBench.]
>
> |Task|SimCLR|MoCo v3|DINO|MAE|SiamMAE|RSP|CropMAE|ToBo     |
> |----|-------|--------|-----|-----|--------|-----|--------|--------|
> |Adroit|40.4±3.3|39.6±4.3|45.6±6.2|39.6±4.3|44.0±6.6|45.6±4.6|50.0±5.1| 60.4±2.2 |
> |MetaWorld|78.4±5.2|65.4±8.0|82.4±5.8|65.4±8.0|81.1±6.3|84.5±6.6|82.4±5.8| 87.8±4.6|
> |DMC|39.7±2.9|43.7±3.2|50.9±1.5|43.7±3.2|56.0±2.9|61.6±3.4|46.4±1.1| 73.5±**0.9** |
> |TriFinger|63.3±3.3|53.3±1.6|64.2±3.5|53.3±1.6|52.1±7.6|66.2±0.8|46.3±1.7|66.5±1.0|
>
> [**Table R8.** Experimental results on vision-based robot policy learning on Franka Kitchen.]
>
> | Tasks | SimCLR | MoCo v3     | DINO    | MAE     | SiamMAE | RSP     | CropMAE   | ToBo     |
> |-------|---------|----------|----------|----------|----------|----------|----------|-----------|
> | Knob1 on      | 25.3±2.1 | 11.5±3.9 | 27.0±3.2 | 12.0±3.3 | 16.8±4.4 | 31.0±2.4 | 31.5±5.3  | 57.3±2.3 |
> | Light on      | 55.8±6.4 | 24.3±5.0 | 44.3±6.5 | 24.3±4.2 | 36.5±7.0 | 44.5±5.6 | 54.0±11.2 | 82.0±1.6 |
> | Sdoor open    | 72.3±2.8 | 66.5±3.2 | 77.0±5.0 | 71.5±4.3 | 68.0±7.9 | 82.5±2.7 | 77.0±8.1  | 95.0±7.1 |
> | Ldoor open    | 17.0±2.9 | 10.3±2.1 | 16.5±2.5 | 12.8±3.9 | 17.3±3.7 | 28.8±4.8 | 25.5±5.7  | 51.0±1.4 |
> | Micro open    | 23.3±2.8 | 14.3±2.5 | 28.5±4.8 | 10.0±2.8 | 13.5±4.8 | 30.3±5.6 | 32.5±4.1  | 55.0±1.4 |
>
>
>
> > ### **W2. Supplement to the limitation part**
>
> Thank you for the advice. We improved the discussion in the limitation part: Due to the resource constraints, we both did not check the scalability of our method beyond huge scale and did not explore beyond the commonly used input resolution of 224x224. Additionally, our study focused on a simplest setting involving two dynamic scenes to learn temporal dynamics. Extending our pipeline to multiple frames setting would be an interesting direction for future work, solving new potential challenges emerging from multi-frame setting.
>
>
> > ### **W3. Extension of ToBo to multi-frame-based approach**
>
> Following the comment, we compare ToBo with the model pre-trained using multiple source frames. Specifically, we randomly sample 4 frames as the source frames and pre-train the model for 100 epochs under the same training recipe to ToBo. As shown in **Table R9**, the multi-frame based model surpasses other baselines (e.g., MAE [1], SiamMAE [2], and RSP [3]) in most of the tasks. However, despite requiring higher pre-training cost, it underperforms compared to ToBo across all robotics tasks. These results suggest that while it is possible to extend ToBo to multi-frame settings, such naive extension may encounter potential new challenges, leading to suboptimal performance.
>
> [**Table R9.** Comparison with the naive extension of ToBo to multi-frame-based setting]
>
> |                             |    Knob1 on   |   Light on    |   Sdoor open    |   Ldoor open    |   Micro open    |
> |------------------------|------------------|-----------------|---------------------|---------------------|---------------------|
> | MAE                |       18.7       |       21.3       |       70.0       |       17.3       |       15.3       |
> | SiamMAE        |       18.0       |       34.0       |       80.7       |       18.7       |       19.3       |
> | RSP                 |       24.7      |       51.7       |       87.3       |       23.3       |       26.7       |
> | |
> | ToBo (multi-frame) |       28.7     |         60.7    |          92.7        |           20.7       |             32.0     |
> | ToBo                      |        **46.7**      |        **78.7**       |        **95.3**          |        **47.3**          |        **37.3**       |
>
>
>
> ### **References**
>
> [1] He et. al., "Masked Autoencoders are Scalable Vision Learners," CVPR 2022.
>
> [2] Gupta et. al., "Siamese masked autoencoders," NeurIPS 2023.
>
> [3] Jang et. al., "Visual representation learning with stochastic frame prediction", ICML 2024.
>
> [4] Gupta et. al., "Relay policy learning: Solving long-horizon tasks via imitation and reinforcement learning. CoRL 2019.
>
> [7] Tassa et. al., "dm_control: Software and tasks for continuous control," arXiv:2006.12983, 2020.
>
> [8] Majumdar et. al., "Where are we in the search for an artificial visual cortex for embodied intelligence?," NeurIPS 2023.
>
> [9] Eymaël et. al., "Efficient image pre-training with siamese cropped masked autoencoders," ECCV 2025.
>
> [10] James et. al., "Rlbench: The robot learning benchmark & learning environment," IEEE Robotics and Automation Letters, 2020.

---

> > ### Comment · Reviewer_GhiU · 2025-08-04
> >
> > Thanks for the authors' response! My questions are resolved. I maintain the previous positive rating.

---

### Official Review · Reviewer_W34j · 2025-07-03

**Clarity:** 2
**Significance:** 2
**Originality:** 2
**Rating:** 4
**Confidence:** 3

**Summary:**

The paper proposes a method for learning visual representations. The method is based on MAE and called Token Bottleneck. The basic idea is to learn representations from a pair of images of the same scene (two different views / video frames). The method trains a model to compress the patches of the first image into a single bottleneck token and, from that token plus visible patches from the second image, predict the masked patches of the second image. The results are evaluated on several simulated robotic tasks and davis label propagation.

**Questions:**

Please see the weaknesses above.

**Ethical Concerns:**

["NO or VERY MINOR ethics concerns only"]

**Final Justification:**

The author rebuttal addressed by questions and updated the score accordingly.

**Limitations:**

Yes.

**Paper Formatting Concerns:**

No formatting concerns.

**Quality:**

2

**Strengths And Weaknesses:**

Strengths:
- I reviewed an earlier version of this paper for ICCV 2025. Compared to that version, the present submission is considerably improved. The writing, method name, and description are much clearer. Some of the missing baselines are included too. I appreciate the updates and thank the authors for incorporating the feedback from the previous review.
- The method is simple, easy to understand, and shows promising signal in tested settings.

Weaknesses:
- 1) Key design choices are not ablated which makes it difficult to understand different aspects of the proposed method. Some examples:
    - a) How many patches to squeeze the information into? Single token vs multiple? How does the tradeoff look like?
    - b) How does the temporal difference impact the performance?
    - c) What would happen if there was no temporal difference between the frames? Would the task be trivial or still useful?
    - d) ...
- 2) It would be good to include comparisons between different methods (e.g., vanilla MAE, SimSiam, etc.) in terms of flops. Both training time flops and inference flops.
- 3) Comparisons on standard benchmarks like ImageNet and Kinetics missing. I understand that the main focus is on simulated robotic evaluation and tracking but it would still be helpful to see how the learnt representations compare to standard approaches like MAE on standard benchmarks.
- 4) It would be good to see straightforward baselines like running vanilla MAE across both views/images.

---

> ### Author Rebuttal · Authors · 2025-07-31
>
> We sincerely thank Reviewer W34j for the thoughtful comments and constructive suggestions. Below, we provide detailed responses to each point.
>
> > ### **W1. Ablation studies on key design choices**
>
>
> We agree that the commented factors are important for understanding the behaviors of ToBo. Thus, we conduct ablation studies on the suggested aspects. All the ablation studies are conducted using models pre-trained for 100 epochs.
>
> * **Ablation study on the number of bottleneck tokens**
>     * We conducted an ablation study on the number of bottleneck tokens, varying it among {1, 2, 4, 8}. As shown in **Table R1**, we observe that using a single token generally yields the best performance across tasks. These results demonstrate that conservative summarization without separate storage is beneficial for understanding the current observation, thereby improving action prediction in robotics
>
>
> * **Ablation study on the impact of temporal difference**
>     * We conducted additional experiments where the maximum frame interval is varied among (48, 96, 144). As shown in **Table R2**, the models are encouraged better to learn to capture dynamic scene evolution when trained with moderate temporal differences, not too short to include meaningful changes and not too long to break temporal coherence.
>
> * **What if there is no temporal difference between the frames?**
>     * Following the comment, we applied our method using the same frame for both the source and target scenes. As shown in **Table R3**, we found that our method still works even without temporal difference, surpassing other baselines (e.g., MAE [1], SiamMAE [2], and RSP [3]) with significant gaps across the tasks. However, its overall performance degrades compared to original ToBo since it loses the opportunity to learn how to capture dynamic evolutions from consecutive scenes. This highlights the importance of temporal contrast for effective pre-training of ToBo.
>
> * **What if multiple source frames are given?**
>     * We further compare ToBo with the model pre-trained using multiple source frames. Specifically, we randomly sample 4 frames as the source frames and pre-train the model for 100 epochs  under the same training recipe to ToBo. As shown in **Table R4**, the multi-frame based model surpasses other baselines (e.g., MAE [1], SiamMAE [2], and RSP [3]) in most of the tasks. However, despite requiring higher pre-training cost, it underperforms compared to ToBo across all robotics tasks. These results suggest that while it is possible to extend ToBo to multi-frame settings, such naive extension may encounter potential new challenges, leading to suboptimal performance.
>
>
>
> [**Table R1.** Ablation study on the number of bottleneck tokens]
>
> |  Number of bottleneck tokens |    Knob1 on   |   Light on    |   Sdoor open    |   Ldoor open    |  Micro open  |
> |----------------------------------------|-------------------|----------------|----------------------|---------------------|------------------|
> | 1                                             |         **46.7**      |        **78.7**       |       **95.3**          |        **47.3**          |        **37.3**       |
> | 2     |         31.0      |        54.0       |        74.0          |        26.0          |        24.0       |
> | 4       |         28.0      |        24.3       |        78.0          |        28.0          |        22.0       |
> | 8     |         10.0     |        20.0       |        56.0          |        26.0          |         9.3       |
>
>
> [**Table R2.** Ablation study on the impact of temporal difference]
>
> | Maximum temporal difference |    Knob1 on   |   Light on    |   Sdoor open    |   Ldoor open    |  Micro open    |
> |--------------------------|-------------------|-----------------|---------------------|---------------------|---------------------|
> | 48                          |         40.7      |        78.7       |        96.0         |        44.0          |        35.3          |
> | 96                          |         **46.7**      |        **78.7**       |        95.3         |        **47.3**          |        37.3          |
> | 144                        |         36.0      |        69.3       |        **97.3**         |        46.7          |        **39.3**          |
>
>
> [**Table R3.** Analysis on the importance of temporal difference]
>
> |                                         |    Knob1 on   |   Light on    |   Sdoor open    |   Ldoor open    |  Micro open    |
> |----------------------------------|------------------|-----------------|---------------------|---------------------|---------------------|
> | MAE                |       18.7       |       21.3       |       70.0       |       17.3       |       15.3       |
> | SiamMAE        |       18.0       |       34.0       |       80.7       |       18.7       |       19.3       |
> | RSP                 |       24.7      |       51.7       |       87.3       |       23.3       |       26.7       |
> | |
> | ToBo (no temporal difference)    |         41.0     |        72.0       |        89.3          |        32.7          |        32.0          |
> | ToBo (with temporal difference)  |         **46.7**      |        **78.7**       |        **95.3**         |        **47.3**          |        **37.3**          |
>
>
> [**Table R4.** Analysis on the multiple source frame]
>
> |                             |    Knob1 on   |   Light on    |   Sdoor open    |   Ldoor open    |   Micro open    |
> |------------------------|------------------|-----------------|---------------------|---------------------|---------------------|
> | MAE                |       18.7       |       21.3       |       70.0       |       17.3       |       15.3       |
> | SiamMAE        |       18.0       |       34.0       |       80.7       |       18.7       |       19.3       |
> | RSP                 |       24.7      |       51.7       |       87.3       |       23.3       |       26.7       |
> | |
> | ToBo (multi-frame) |       28.7     |         60.7    |          92.7        |           20.7       |             32.0     |
> | ToBo                      |        **46.7**      |        **78.7**       |        **95.3**          |        **47.3**          |        **37.3**       |
>
>
> > ### **W2. Comparison of training and inference flops**
>
> We conducted FLOPs evaluation for both training and inference to quantitatively compare the computational cost of each model, as summarized in **Table R5**.
> * During **inference**, all models use the same backbone architecture and input resolution without any input masking, resulting in identical inference FLOPs at the same model scale (e.g., 4.6 GFLOPs for ViT-Small).
> * During **training**, ToBo, MAE [1], and SiamMAE [2] show similar computational costs while RSP [3] requires substantially more computation of 32.5 GFLOPs due to its complex decoding mechanisms. When considering computational costs with downstream performance (e.g., performance in Franka Kitchen [4]), these results further support the effectiveness of ToBo, which achieves a strong balance between efficiency and performance.
>
> [**Table R5.** Comparison of training and inference FLOPs and downstream performance in Franka Kitchen ]
>
> | Method    | Training FLOPs (GFLOPs) | Inference FLOPs (GFLOPs) | Franka Kitchen (%) |
> |-----------|-------------------------|---------------------------|---------------------|
> | MAE      | 13.0                    | 4.6                       | 26.1                |
> | SiamMAE  | 13.1                    | 4.6                       | 30.4                |
> | RSP     | 32.5                    | 4.6                       | 43.4                |
> | |
> | ToBo      | 15.9                    | 4.6                       | 68.1                |
>
>
> > ### **W3. Comparisons of ToBo and MAE on Kinetics-400**
>
> Following the comment, we compare the action classification accuracy of ToBo and the baselines (i.e., MAE [1], SiamMAE [2], and RSP [3]) on the Kinetics-400 dataset [6]. To this end, we fine-tune all models under a reduced setting of 40 epochs, due to the limited rebuttal period. As shown in **Table R6**, while the performance gaps are relatively smaller than those appears in our main downstream tasks in the main paper, ToBo achieves the highest accuracy among all models, demonstrating its effectiveness on this standard video classification benchmark.
>
> [**Table R6.** Comparison of ToBo and other baselines in action recognition in Kinetics-400]
>
> |                  |   Kinetics-400 (%)   |
> |---------------|-----------------|
> | MAE     |  39.9          |
> | SiamMAE  |  38.1          |
> | RSP           | 38.2           |
> | |
> | ToBo         |    **40.8**         |
>
>
>
> > ### **W4. Comparison with straightforward baselines**
>
> We apologize for the confusion. We would like to clarify that MAE [1] in the tables of the main paper is indeed a straightforward baseline that performs masked image modeling independently on both input views. We will improve the description for the baselines.
>
>
> ### **References**
>
> [1] He et. al., "Masked Autoencoders are Scalable Vision Learners," CVPR 2022.
>
> [2] Gupta et. al., "Siamese masked autoencoders," NeurIPS 2023.
>
> [3] Jang et. al., "Visual representation learning with stochastic frame prediction", ICML 2024.
>
> [4] Gupta et. al., "Relay policy learning: Solving long-horizon tasks via imitation and reinforcement learning. CoRL 2019.
>
> [5] Russakovsky et. al., "Imagenet large scale visual recognition challenge," IJCV 2015.
>
> [6] Kay et. al., "The kinetics human action video dataset," arXiv:1705.06950, 2017.

---

> > ### Comment · Area_Chair_NAu8 · 2025-08-05
> > **Were the comments addressed?**
> >
> > Dear Reviewer W34j,
> >
> > The authors have provided extensive ablations on design choices,  flops and other datasets. As the Author-Reviewer discussion is coming to an end, we appreciate your participation in the discussion. Did the the authors answer your concerns? Does it change your score?
> >
> > -- Your AC

---

> > ### Comment · Reviewer_W34j · 2025-08-05
> > **Reviewer comment**
> >
> > I thank the authors for the response. The rebuttal addressed my questions.

---

> ### Author Response · Authors · 2025-08-06
>
> Thank you again for your constructive comments! We're glad that your questions have been addressed. Are there any remaining points we can further clarify? Please feel free to let us know if there are any remaining concerns!

---

### Note · Authors · 2025-08-14

We sincerely thank the SAC, AC, and reviewers for their efforts and valuable feedback throughout the review process.
We are grateful that our work has been commended for the following aspects:
* **Promising results** (W34j, GhiU, dztw)
* **Simple method** (W34j, GhiU)
* **Results with error bars** (GhiU)
* **Novel method** (dztw)
* **Extensive experiments** (dztw)
* **Well-written** (3Edo)
* **Interesting insight** (3Edo)


Following the reviewers' comments, we provided additional analyses and clarifications:
* Comparisons of training and inference flops (W34j)
* Comparisons on standard benchmarks (W34j)
* Comparison with DINO v2, I-JEPA, V-JEPA, i-BOT, T-CORE (dztw, 3Edo)
* Ablation studies on key design choices (W34j, dztw)
* Extension of ToBo to multi-frame-based approach (GhiU)
* Scalability of our method when trained on more samples (dztw)
* Clarification for straightforward baselines  (W34j)
* Clarification for large variances (GhiU)
* Clarification for much smaller improvements on label propagation (dztw)
* Clarification on the design of the decoder architecture (3Edo)
* Supplement to the limitation part (GhiU)

We are pleased that all the reviewers confirmed their concerns were addressed.
Finally, we would like to revisit our core insight that Token Bottleneck conservatively summarizes observed scenes in a way that also effectively embeds temporal dynamics. Extensive experiments have demonstrated the effectiveness of Token Bottleneck across various downstream tasks. We believe that our research makes a meaningful contribution and will inspire promising methods in the future.

---

### Decision · Program_Chairs · 2025-09-17

**Decision:**

Accept (poster)

**Comment:**

The paper introduces a Token Bottleneck approach that allows to learn a self-supervised representation of the sequential scene by squeezing it into a token bottleneck. The model is trained to reconstruct a scene solely from the token bottleneck, together with a few target “hints” (patches of the target scene). The usefulness of the approach is demonstrated on real-world robot manipulation tasks.

The reviewers praise that the approach is simple but effective, particularly performing well on the robotics tasks. During the rebuttal, the authors also extended the ToBo method to a multi-frame approach, which added a valuable comparison. The authors also provided extensive comparisons and ablations to other existing self-supervised methods and demonstrated that token bottleneck remains superior.

The recommendation is Accept.